# Working with Children with ADHD—A Latent Profile Analysis of Teachers' and Psychotherapists' Attitudes

**Martina Dort** [1],* **, Anna Enrica Strelow** [1] **, Malte Schwinger** [2] and **Hanna Christiansen** [1]

[1] Clinical Child and Adolescent Psychology, Department of Psychology, University of Marburg, 35032 Marburg, Germany; anna.strelow@uni-marburg.de (A.E.S.); hanna.christiansen@uni-marburg.de (H.C.)

[2] Educational Psychology, Department of Psychology, University of Marburg, 35032 Marburg, Germany; malte.schwinger@uni-marburg.de

* Correspondence: martina.dort@uni-marburg.de

**Abstract:** A positive attitude of teachers and psychotherapists towards children with ADHD can both support their mutual relationship and support reducing ADHD-related symptoms. According to Fishbein and Ajzen's rational-choice approach, attitude formation is based on a person's expectations and the appraisal of these, thus attitude, therefore, differs individually. The present study aimed to identify different attitude profiles based on our participants' answer patterns on the ADHD-school-expectation questionnaire's (ASE) subscales, and to examine which attitude profile would be desirable for professionals working with children with ADHD. We conducted a latent profile analysis and investigated differences between the latent profiles. Our analysis revealed three attitude profiles characterized by negative, moderate and extreme ratings of expectations. The attitude profiles differed in further variables such as the use and effectiveness of rating classroom management strategies, knowledge of ADHD, perceived control, stress and strain, as well as some personality traits. The extreme rating profile seems to be beneficial for children with ADHD, whereas the moderate rating profile might appeal to certain professionals.

**Keywords:** ADHD; attitude; teachers; psychotherapists; latent profiles; classroom management strategies

## 1. Introduction

Statistically speaking, one to two children worldwide in every classroom (with an average class size of 30 children) will suffer from Attention Deficit/Hyperactivity Disorder (ADHD) [1,2]. The core symptoms—inattention, hyperactivity, and impulsivity—appear in the classroom through behaviors such as not listening, not following instructions, fidgeting in the chair, or blurting out answers—behavioral problems that contribute to perceived classroom stress. Thus, ADHD is often first identified when families seek professional support (e.g., psychotherapy, psychiatric care) once children have entered school [3]. Hence, both teachers and clinicians are professionals often in contact with children with ADHD, and whose encounters with such children may not be always positive. An important factor for such contacts is to opt instead for solid, supportive collaboration, thus improving the relationship between the professionals and children with ADHD [4,5].

Studies demonstrate that teachers' behavior influences the work habits of children with ADHD positively, as do social processes in the classroom such as the teachers' feedback, which affects their peers' perception of a child [6–8]. If teachers apply specific classroom management strategies (CMS), their behavior can also significantly change children's behavior, for instance by leading to reduced ADHD symptoms [9]. Such use of CMS can contribute to both a less stressful classroom environment and potentially disrupt dysfunctional trajectories, as handling ADHD-related symptoms positively

might affect developmental pathways and result in disrupting certain psychopathologies [10–13]. Further, teachers can influence a student's performance by their expectations about that student—even when teachers are trying to seem neutral [14].

However, students with ADHD often feel misunderstood and treated unfairly by their teachers, whereas teachers often feel overwhelmed by the children's behavior, resulting in an uncomfortable situation for both and a worsening relationship [15,16]. Such bilateral discomfort might originate from mutual attitudes, as attitudes constitute a person's expectations and their ratings (i.e., positive or negative) thereof [17]. It is assumed that our cognitive (beliefs) and affective attitude (expected elicited emotions) towards a person influence our behavioral attitude [18,19] which, according to the Theory of Planned Behavior (TPB) [20], influences our intention to engage in a specific behavior that finally results in that behavior being shown. With respect to teachers working with children with ADHD, their potentially negative attitudes might contribute to an interaction spiral so that they repudiate children with ADHD who, in turn, keep acting out, as they are feeling misunderstood and rejected [18].

Psychotherapists are another group that regularly works with children with ADHD, thus they also form attitudes that result in specific interactive behavior with those children [18–20]. In contrast to teachers, however, there is, to our knowledge, no literature to date examining the attitude of psychotherapists towards children with ADHD. As psychotherapists are professionals who have explicitly chosen to work with patients exhibiting behavior problems, one could hypothesize that in general, they would be understanding and show a positive attitude towards children with ADHD. On the other hand, such children might be perceived as demanding and difficult, and psychotherapists might prefer patients with other disorders to those with ADHD.

As both teachers and psychotherapists are professionals working with children, and as attitudes towards children with ADHD can influence their interactions with such children, this study aims to identify latent attitude profiles of both groups. We assume that some profiles might prove to be more advantageous than others when working with children with ADHD. For this purpose, we also consider direct experiences, social influences, and individual differences that are known to influence expectations and, consequently, attitudes [17,21,22]. Especially in the case of teachers, those factors might also influence the use of CMS [23,24].

As Strelow, Dort, Schwinger, and Christiansen point out [25], an important *direct experience* in this context can be the perceived stress that children with ADHD elicit that supports the expectation of further negative interactions. Job experience can also be assumed to be a direct experience with potential influence on expectations and thus attitude. However, previous studies found no effect of job experience of in-service teachers on their intention to use effective CMS and on their attitude towards children with ADHD; only an effect moderated via knowledge and perceived control [26,27]. Next to in-service teachers, a big group of our sample was pre-service teachers that cannot be expected to have job experience yet. Therefore, it was not of additional value to assess this variable in this group. Accordingly, we did not take this variable into account in the present study. The age of the children professionals work with could also shape experiences, as studies show different handling of ADHD symptoms of children in primary- and secondary school [27–29]. Due to the fact that the different (sub) groups in this sample work with children with a different range of ages, a clear categorization of the children's age would have been difficult. Thus, the present study aims to give a general overview of professionals' attitude towards children with ADHD.

Considering social influences, the subjective norm (i.e., what I perceive as relevant in a specific context) could be relevant, as according to the TPB, this variable influences both the formation of expectations and the intention to display a specific behavior, such as the use of CMS.

Perceived behavioral control should not just be considered as an individual difference influencing expectations, as it also influences the intention to reveal a specific behavior and its actual realization [20]. Knowledge about ADHD and general stress represent further individual differences known to influence teachers' expectations and their attitude toward children with ADHD [30–32]. Additionally, study results suggest the Big Five personality traits, as well as Social Dominance

Orientation (SDO) and Right Wing Authoritarianism (RWA), play an important role in predicting individual prejudices that influence expectations and attitudes [22,33]. The disposition stress reactivity may also be relevant in this context, as it moderates the relationship between the stress event and stress reaction, which, in turn, are related to the concept of abilities [34,35], and therefore perceived behavioral control. Stress reactivity could thus influence the aforementioned potentially relevant factors, namely strain, perceived stress elicited by children with ADHD, and perceived behavioral control.

To summarize, we aim to (1) compare the attitudes of teachers and psychotherapists towards children with ADHD; (2) identify latent attitude profiles; and (3) investigate which attitude profile is advantageous for professionals working with children with ADHD by taking the potential moderators direct experiences, social influences, and individual differences into account. We further hypothesize that psychotherapists' attitude will be more positive than teachers' attitude and that latent attitude profiles will differ in the variables stress, perceived control, subjective norm, knowledge as well as personality traits.

## 2. Materials and Methods

### 2.1. Study Design and Procedure

This study was conducted via three online surveys on https://www.soscisurvey.de addressing pre-service, who we defined as education majors that are not regularly teaching in schools, and in-service teachers as well as psychotherapists in training (PIT). The surveys differed only in the wording, which was adjusted for each group according to its usage in their working context (e.g., pupils for teachers—children for psychotherapists). Pre-service teachers and PIT were assumed to be rather accessible groups that reflect the effects of the current education of professionals that work with children with ADHD. In-service teachers also represent a big group that works with children with ADHD. Additionally, this group can be assumed to have quite a lot of experience in working with affected children. Therefore, this group was also included in the study.

A cover letter provided detailed study information as well as the corresponding link to the online survey that was disseminated via different German university e-mail lists, schools, institutions for psychotherapy training and Facebook groups for pre- and in-service teachers. Data collection lasted, on average, five weeks, including a reminder after the first three weeks. Each group had the option of choosing a Nintendo Switch, two tickets to a musical, or a spa weekend for two (value about 350 EUR); three vouchers with a total value of 150 EUR were raffled to encourage participation in the study.

### 2.2. Participants

We collected data from N = 1794 participants. Detailed information on participants' characteristics are presented in Table 1.

### 2.3. Measures

#### 2.3.1. The ADHD School Expectation Questionnaire (ASE)

Attitude towards children with ADHD, knowledge about ADHD, and the use of and attitude towards interventions for children with ADHD were measured via the ADHD-school-expectation questionnaire (ASE) [36]. It assesses attitude with 33 items. For every item, the expectation from 0 = unlikely to 1 = likely and the related rating from −3 = negative to 3 = positive are stated on separate visual analogue scales (VAS). The variable attitude is then calculated by the multiplication of expectation and its related rating for every item, which are added together to obtain a total scale value, although two subscales (attitude towards positive aspects and attitude towards negative aspects) can be derived [17]. Cronbach's α calculation for the total attitude scale is 0.85.

**Table 1.** Detailed information about N = 1794 participants' characteristics.

| Group | N | % | Male | Female | Diverse | Age *M (SD)* |
|---|---|---|---|---|---|---|
| pre-service teachers | 1086 | 60.5 | 332 | 749 | 5 | 23.22 (3.93) |
| PIT | 109 | 6.1 | 21 | 88 | 0 | 30.94 (5.24) |
| in-service teachers | 599 | 33.4 | 106 | 493 | 0 | 41.33 (10.01) |
| Total | 1794 | 100.0 | 459 | 1330 | 5 | 29.74 (10.73) |
| **Subgroup** | | | | | | |
| pre-service_elementary school | 279 | 15.6 | 44 | 234 | 1 | 22.54 (3.92) |
| pre-service_middle school | 179 | 10.0 | 64 | 114 | 1 | 24.08 (4.48) |
| pre-service_senior high school | 488 | 27.2 | 193 | 293 | 2 | 23.16 (3.59) |
| pre-service_special needs school | 109 | 6.1 | 22 | 87 | 0 | 23.31 (3.70) |
| in-service_elementary school | 304 | 16.9 | 30 | 274 | 0 | 42.23 (9.81) |
| in-service_middle school | 157 | 8.8 | 45 | 112 | 0 | 39.52 (9.37) |
| in-service_senior high school | 42 | 2.3 | 16 | 26 | 0 | 38.24 (9.85) |
| in-service_special needs school | 56 | 3.1 | 10 | 46 | 0 | 43.59 (11.04) |
| PIT—children | 105 | 5.9 | 19 | 86 | 0 | 30.96 (5.33) |
| PIT—adults | 4 | 0.2 | 2 | 2 | 0 | 30.50 (2.52) |

Note: PIT = psychotherapist in training; subgroup contains *n* = 71 missing.

The knowledge scale in the ASE contains 24 items about symptoms, etiology, diagnostics & prevalence, and interventions. All items are answered with a VAS from true to false. A correct answer within the first sixth of the VAS is granted one knowledge point; Cronbach's $\alpha$ is 0.81.

The intervention scale in the ASE includes 15 effective and 12 ineffective intervention strategies that are rated on two VAS according to their usage from 0 = never to 1 = very often and the estimated effectiveness from 0 = not effective at all to 1 = very effective. For the present data, Cronbach's $\alpha$ was 0.72 for the usage and 0.73 for the effectiveness rating.

### 2.3.2. Perceived Attitude towards Children with ADHD

Participants' perceived attitude was assessed by requesting them to state how positive or negative they judged their attitude to be towards children with ADHD. This was answered on a VAS from −3 = negative to 3 = positive [25,36].

### 2.3.3. Perceived Behavioral Control

Perceived behavioral control was measured with the two items I have the ability to teach children with ADHD effectively and Dealing with children with ADHD exceeds my abilities constructed according to the TPB [20]. The answering format was a VAS varying from 0 = totally disagree to 5 = totally agree [25,36]. Cronbach's $\alpha$ was 0.82 for the present data.

### 2.3.4. Subjective Norm

Subjective norm was measured with three items: I want persons who are important to me to think positively about me; To do something that I know others consider to be unethical makes me lose my self-respect; I don't care whether others have a poor opinion about me. The items were assessed with a VAS ranging from 0 = totally disagree to 5 = totally agree [25,36]. Cronbach's $\alpha$ for the present data was 0.51.

### 2.3.5. Personality

To assess personality, we used the short German version of the Big Five Inventory (BFI) [37]. It contains 21 items and uses a five-point Likert-scale ranging from 0 = very inapplicable to

5 = very applicable to assess the facets Extraversion, Agreeableness, Conscientiousness, Neuroticism, and Openness. For the present data, Cronbach's $\alpha$ calculation revealed the following results: Extraversion $\alpha$ = 0.78, Agreeableness $\alpha$ = 0.57, Conscientiousness $\alpha$ = 0.74, Neuroticism $\alpha$ = 0.68, and Openness $\alpha$ = 0.76.

### 2.3.6. Social Dominance Orientation (SDO)

SDO was measured via the German scale designed by Cohrs, Moschner, Maes, and Kielmann [38], based on the scale by Pratto, Sidanius, Stallworth, and Malle [39] and Six, Wolfrath, and Zick [40]. The 12 items (e.g., Social equality should increase.) are answered on a six-point Likert-scale ranging from 0 = I totally disagree to 6 = I totally agree. Cronbach's $\alpha$ was 0.80 for the present data.

### 2.3.7. Right-Wing Authoritarianism (RWA)

RWA was measured via the German short-scale *Kurzskala Autoritarismus* (KSA-3) by Beierlein, Asbrock, Kauff, and Schmidt [41]. Three items (e.g., We need strong leaders to live secure in society) was answered applying a six-point Likert-scale ranging from 0 = I totally disagree to 6 = I totally agree. Cronbach's $\alpha$ calculation was 0.58 for the present data.

### 2.3.8. Stress Reactivity

The Perceived Stress Reactivity Scale (PSRS) by Schlotz, Yim, Zoccola, Jansen, and Schulz [42] was used to measure stress reactivity. It contains 23 items, each representing the first part of a statement (e.g., If I have done something wrong . . . ) that is completed with one of three answering options ( . . . I generally keep my self-confidence/I sometimes become insecure about my abilities/I often question my abilities). Apart from the total scale, five subscales can be calculated. For the present data, Cronbach's $\alpha$ was 0.87.

### 2.3.9. Strain

Psychological strain was measured with the short version of the Brief Symptom Inventory (BSI) [43]. The comprehensive Global Severity Index (GSI) based on 18 items was used. A four-point Likert-scale ranging from 0 = nothing at all to 4 = very strong was used. Cronbach's $\alpha$ was 0.81 for the GSI.

### 2.3.10. Perceived Stress Elicited by Children with ADHD

Perceived stress elicited by children with ADHD was measured with the question How severe do you find your stress to be due to the behavior of children with ADHD? That was answered on a VAS ranging from 0 = not severe at all to 6 = very severe

### *2.4. Data Analysis*

Data editing (excluding surveys with less than 75% answers per scale, recoding items and calculating scale scores) was followed by descriptive statistical analyses. Subsequently, inference statistical analyses were conducted. Differences in the attitude values towards children with ADHD measured with the ASE and via self-assessment between the three groups pre-service teachers, in-service teachers and PIT were measured via a multivariate analysis of variance (MANOVA).

### Latent Profile Analysis

We performed a latent profile analysis (LPA) to identify latent classes of participants who showed a similar response pattern with respect to attitudes towards children with ADHD. Therefore, we used participants' answers on the attitude subscales expectation of positive aspects, expectation of negative aspects, rating of positive aspects, and rating of negative aspects as class indicators. For the LPA, we used z-standardized data due to the different scale formats of the measured latent class indicators.

The default settings of Mplus 8.4 (2019) indicating free estimation of means and variances of the class indicators, constant variances in class indicators between classes, and a covariance set to 0 between indicators within one class were kept. The LPA was conducted for two to eight class solutions, the latter representing high and low values of the examined parameters. The selection of the best class solution was based on the Akaike Information Criterion (AIC), Bayesian Information Criterion (BIC) and sample size-adjusted BIC (ssaBIC), whereupon the class solution with the lowest values should be chosen. In addition, the Lo–Mendell–Rubin likelihood ratio test of model fit (LMR), the Vuong–Lo–Mendell–Rubin likelihood ratio test (VLMRT) [44] and the parametric bootstrapped LRT (BLRT) [45] were taken into account. Considering these tests, the class solution with significant *p*-values should be chosen, as those tests measure whether adding a class improves or worsens the total solution [46]. The entropy value was also included in the choice of a class solution with higher values, indicating more precise assignment of participants to latent classes [47]. The best fitting models were finally compared with respect to the interpretability of their profile structure [48,49].

In the next step, identified profiles were related to different correlates. There are different ways of validly assessing the effects of latent classes on such "distal outcomes". Here, we used the automatic version of the BCH method, in which an ANOVA weighted by the inverse classification error probabilities is calculated [50], to estimate the means of the outcome variables across the different classes [51].

## 3. Results

### 3.1. Descriptive Statistics and MANOVA

The descriptive results of the measured variables for all groups are presented in Table 2. The MANOVA revealed a significant difference in attitudes measured with the ASE between the groups of pre-service teachers, in-service teachers and PIT, $F_{(2, 1768)} = 4.936$, $p = 0.007$. Post-hoc tests showed that PIT had significantly more positive attitudes towards children with ADHD ($M = -8.84$, $SD = 14.45$) than in-service teachers ($M = -13.12$, $SD = 14.66$), $p = 0.012$, Cohen's d = 0.294. The MANOVA for the self-assessed attitude also resulted in a significant group difference, $F_{(2, 1768)} = 5.678$, $p = 0.003$. In this case, post-hoc tests revealed a significantly more negatively perceived attitude of PIT ($M = -0.52$, $SD = 1.65$) than in pre-service teachers ($M = -0.03$, $SD = 1.66$), $p = 0.008$, Cohen's d = $-0.296$.

### 3.2. Latent Profile Analysis

The comparison of different LPA solutions indicated a solution with three latent attitude classes as best, as solutions with more classes did not result in significant VLMRT and LMR values while the AIC, BIC and ssaBIC values kept falling. Moreover, the entropy value dropped from the four-class-solution further. Compared to the two-class-solution, the three-class-solution was more differentiated, with profiles that were easier to interpret. The fit indices and class counts for the two- to four-class solution are presented in Table 3. For the three-class solution, the *z*-standardized and original means of the attitude scales are presented in Table 4. The latent profiles are illustrated in Figure 1. Furthermore, the three latent classes' demographics are presented in Table 5. The attitude profiles differed mainly in how positive and negative aspects were rated. Our results suggest that 22% of the participants have a rather *negative rating profile* that is mainly influenced by a negative rating of negative aspects and a negative rating of positive aspects. People in this profile also had the most negative total attitude score compared to the other profiles ($M = 25.80$, $SD = 11.15$). The class with a *moderate rating profile* contained 27% of the participants in our entire sample and obtained the most positive total attitude score ($M = -3.18$, $SD = 11.05$) compared to the other profiles. About half of the participants (52%) showed a rather *extreme rating profile* (total attitude score $M = -10.65$, $SD = 12.46$) with extreme positive and extreme negative ratings.

**Table 2.** Means and standard deviations of all assessed variables for N = 1794 participants.

| Group | | Attitude | Perceived Attitude | Knowledge | Use of CMS Total | Use of Effective CMS | Use of Ineffective CMS | Rating Of CMS Total | Rating of Effective CMS | Rating of Ineffective CMS | Perceived Behavioral Control | Subjective Norm |
|---|---|---|---|---|---|---|---|---|---|---|---|---|
| pre-service teachers | M | −11.60 | −0.03 | 7.23 | 0.55 | 0.72 | 0.35 | 0.52 | 0.73 | 0.26 | 2.60 | 4.65 |
| | SD | 13.92 | 1.66 | 4.21 | 0.09 | 0.12 | 0.14 | 0.09 | 0.13 | 0.12 | 1.28 | 22.50 |
| in-service teachers | M | −13.12 | −0.20 | 9.25 | 0.57 | 0.74 | 0.36 | 0.54 | 0.76 | 0.26 | 2.82 | 11.03 |
| | SD | 14.66 | 1.68 | 4.29 | 0.09 | 0.13 | 0.14 | 0.09 | 0.14 | 0.13 | 1.24 | 50.22 |
| PIT | M | −8.84 | −0.52 | 12.57 | 0.52 | 0.81 | 0.16 | 0.53 | 0.82 | 0.17 | 3.80 | 6.39 |
| | SD | 14.45 | 1.65 | 3.73 | 0.09 | 0.11 | 0.12 | 0.07 | 0.09 | 0.10 | 0.94 | 31.59 |
| Total | M | −11.93 | −0.12 | 8.23 | 0.56 | 0.73 | 0.34 | 0.53 | 0.75 | 0.26 | 2.74 | 6.89 |
| | SD | 14.23 | 1.67 | 4.45 | 0.09 | 0.13 | 0.15 | 0.09 | 0.13 | 0.13 | 1.28 | 34.88 |

| Group | | Extraversion | Agreeableness | Conscientiousness | Neuroticism | Openness | SDO | RWA | Stress Reactivity | Strain | Perceived Stress Elicited by Children with ADHD |
|---|---|---|---|---|---|---|---|---|---|---|---|
| pre-service teachers | M | 3.67 | 3.46 | 3.71 | 2.82 | 3.86 | 1.12 | 2.08 | 20.74 | 3.75 | 3.18 |
| | SD | 0.71 | 0.58 | 0.66 | 0.73 | 0.71 | 0.65 | 0.90 | 7.40 | 0.73 | 0.98 |
| in-service teachers | M | 3.89 | 3.73 | 3.98 | 2.60 | 4.00 | 1.16 | 2.09 | 20.79 | 3.57 | 3.28 |
| | SD | 0.65 | 0.52 | 0.61 | 0.67 | 0.65 | 0.65 | 0.94 | 7.46 | 0.63 | 1.25 |
| PIT | M | 3.71 | 3.83 | 3.98 | 2.62 | 3.97 | 0.99 | 1.65 | 19.31 | 3.50 | 4.11 |
| | SD | 0.72 | 0.43 | 0.57 | 0.63 | 0.61 | 0.57 | 0.78 | 6.04 | 0.41 | 0.67 |
| Total | M | 3.75 | 3.57 | 3.81 | 2.74 | 3.91 | 1.13 | 2.06 | 20.67 | 3.67 | 3.27 |
| | SD | 0.70 | 0.57 | 0.65 | 0.71 | 0.69 | 0.64 | 0.91 | 7.35 | 0.69 | 1.09 |

Note: PIT = psychotherapist in training, CMS = classroom management strategies, SDO = social dominance orientation, RWA = right-wing authoritarianism.

**Table 3.** LPA results based on *N* = 1794 participants' answers on the ASE's attitude subscales expectation of positive aspects, expectation of negative aspects, rating of positive aspects, and rating of negative aspects. Presented are the fit indices as well as the class counts and proportions based on their most likely latent class membership. The best solution is presented in bold.

| N$_{classes}$ | LogL | AIC | BIC | ssaBIC | VLMRT | LMR | BLRT | Entropy | Class Counts and Proportions | | | |
|---|---|---|---|---|---|---|---|---|---|---|---|---|
| | | | | | | | | | Class 1 | Class 2 | Class 3 | Class 4 |
| 1 | −10,134.896 | 20,285.793 | 20,329.730 | 20,304.315 | - | - | - | - | 1794 (100%) | | | |
| 2 | −9823.516 | 19,673.032 | 19,744.431 | 19,703.131 | <0.01 | <0.01 | <0.01 | 0.855 | 1042 (58%) | 752 (42%) | | |
| **3** | **−9399.986** | **18,835.973** | **18,934.833** | **18,877.648** | **0.002** | **0.003** | **<0.01** | **0.855** | **388 (22%)** | **480 (27%)** | **926 (52%)** | |
| 4 | −9239.130 | 18,524.261 | 18,650.581 | 18,577.512 | 0.450 | 0.455 | <0.01 | 0.835 | 519 (29%) | 310 (17%) | 147 (8%) | 818 (46%) |

Note: LogL = Log Likelihood; AIC = Akaike Information Criterion; BIC = Bayesian Information Criterion; ssaBIC = sample size-adjusted BIC; LMR = Lo-Mendell-Rubin likelihood ratio test of model fit; VLMRT = Vuong-Lo-Mendell-Rubin likelihood ratio test; BLRT = parametric bootstrapped LRT.

**Table 4.** LPA three-class-solution. z-standardized and original means of the ASE's attitude subscales for *N* = 1794.

| Class | *n* | Expectation of Positive Aspects | | Expectation of Negative Aspects | | Rating of Positive Aspects | | Rating of Negative Aspects | |
|---|---|---|---|---|---|---|---|---|---|
| | | z-Values *M (SD)* | Original Values *M (SD)* | z-Values *M (SD)* | Original Values *M (SD)* | z-Values *M (SD)* | Original Values *M (SD)* | z-Values *M (SD)* | Original Values *M (SD)* |
| negative | 388 | −0.605 (0.833) | 0.300 (0.101) | 0.412 (0.893) | 0.754 (0.100) | −1.429 (0.161) | −1.003 (0.629) | −0.244 (0.639) | −1.816 (0.611) |
| moderate | 480 | 0.564 (0.833) | 0.457 (0.101) | −0.492 (0.893) | 0.646 (0.106) | −0.416 (0.161) | 0.522 (0.642) | 0.975 (0.639) | −0.861 (0.704) |
| extreme | 926 | −0.046 (0.833) | 0.374 (0.128) | 0.088 (0.893) | 0.714 (0.113) | 0.826 (0.161) | 2.381 (0.486) | −0.416 (0.639) | −1.936 (0.524) |

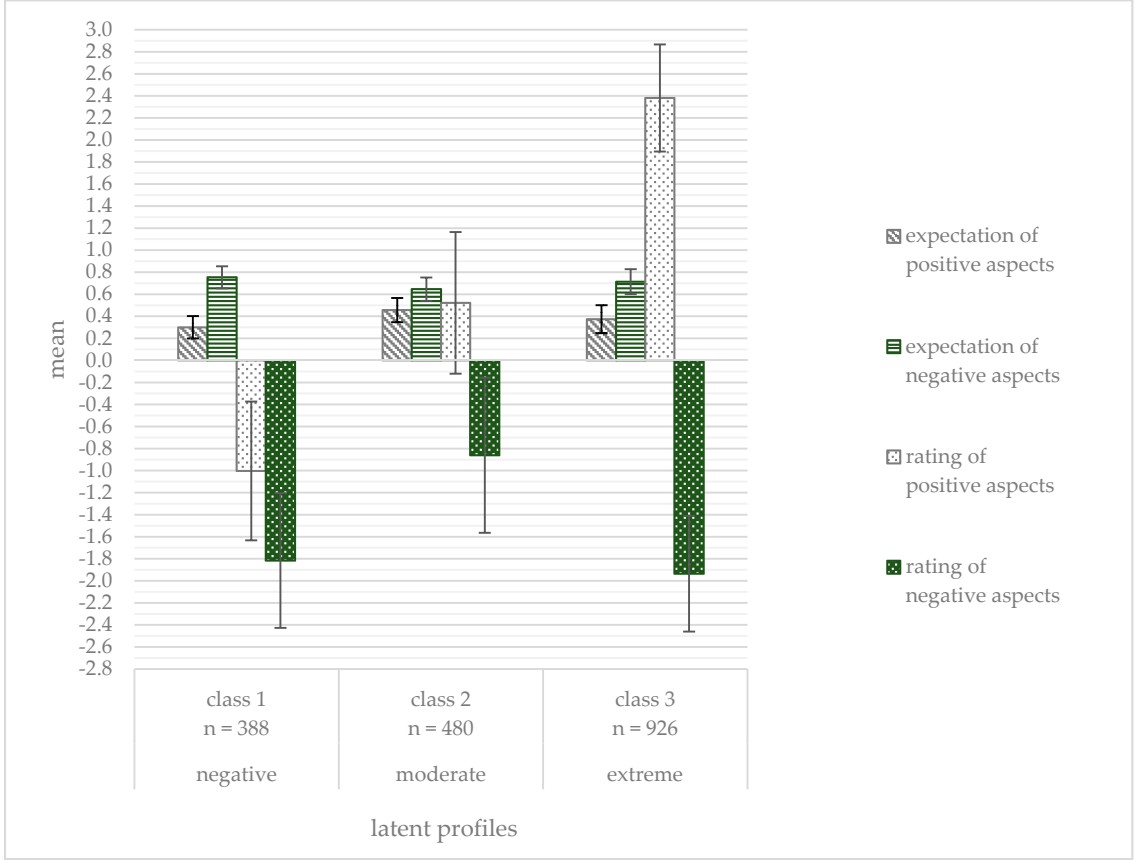

**Figure 1.** Latent profiles based on N = 1794 participants' answers on the ASE's attitude subscales *expectation of positive aspects*, *expectation of negative aspects*, *rating of positive aspects*, and *rating of negative aspects*. Latent profile analysis was conducted using z-standardized values. Means of the original values are presented to ease comprehension.

**Table 5.** LPA three-class-solution. Demographics for N = 1794: the proportion of each group (pre-service teachers, in-service teachers or PIT) in a class is shown in brackets first, followed by the proportion of class members compared to the whole group of pre-service teachers, in-service teachers or PIT.

| Class | Age M (SD) | Male | Female | Diverse | Pre-Service Teachers | in-Service Teachers | PIT |
|---|---|---|---|---|---|---|---|
| negative | 31.48 (12.10) | 24% | 76% | 0.3% | 210 (54%/19%) | 159 (41%/27%) | 19 (5%/17%) |
| moderate | 30.45 (11.02) | 30% | 70% | 0% | 290 (60%/27%) | 163 (34%/27%) | 27 (6%/25%) |
| extreme | 28.64 (9.81) | 24% | 76% | 0.4% | 586 (63%/54%) | 277 (30%/46%) | 63 (7%/58%) |

Note: PIT = psychotherapists in training.

Aside from their attitude profiles, the latent classes also differed significantly in several other variables. In Table 6, we listed the significant differences according to the BCH-method. For better traceability, the original values are reported. Corresponding z-values are found in the Supplementary Material.

**Table 6.** LPA three-class-solution. Significant differences in auxiliary variables between classes according to the BCH method. Presented are original means and standard deviations for the variables for $N$ = 1794. In a second row corresponding Cohen's |*d*|values for significant group differences are presented.

| Class | Perceived Attitude | Knowledge | Perceived Behavioral Control | Perceived Stress | Use Ineffective CMS | Rating Effective CMS | Rating Ineffective CMS | SDO | RWA | Stress Reactivity | Extra-Version |
|---|---|---|---|---|---|---|---|---|---|---|---|
| negative [a] | −0.57 (1.67) [bc] | 8.65 (4.43) [b] | 2.57 (1.32) [b] | 3.53 (1.07) [bc] | 0.37 (0.16) [bc] | 0.72 (0.15) [bc] | 0.26 (0.14) [bc] | 1.23 (0.68) [c] | 2.24 (0.99) [bc] | 21.56 (8.05) [b] | 3.81 (0.71) [c] |
| moderate [b] | 0.51 (1.56) [ac] | 7.17 (4.56) [ac] | 3.00 (1.18) [ac] | 3.01 (1.08) [ac] | 0.33 (0.14) [a] | 0.74 (0.13) [ac] | 0.28 (0.14) [ac] | 1.18 (0.64) [c] | 1.97 (0.93) [a] | 19.51 (7.15) [ac] | 3.79 (0.69) [c] |
| extreme [c] | −0.25 (1.63) [ab] | 8.59 (4.37) [b] | 2.67 (1.29) [b] | 3.30 (1.07) [ab] | 0.34 (0.14) [a] | 0.76 (0.12) [ab] | 0.24 (0.12) [ab] | 1.06 (0.62) [ab] | 2.03 (0.86) [a] | 20.90 (7.08) [b] | 3.69 (0.69) [ab] |

| Class | \multicolumn Perceived Attitude | | | Knowledge | | | Perceived Behavioral Control | | | Perceived Stress | | | Use Ineffective CMS | | | Rating Effective CMS | | | Rating Ineffective CMS | | | SDO | | | RWA | | | Stress Reactivity | | | Extra-Version | | |
|---|---|---|---|---|---|---|---|---|---|---|---|---|---|---|---|---|---|---|---|---|---|---|---|---|---|---|---|---|---|---|---|---|---|
| | a | b | c | a | b | c | a | b | c | a | b | c | a | b | c | a | b | c | a | b | c | a | b | c | a | b | c | a | b | c | a | b | c |
| negative [a] | - | 0.7 | 0.2 | - | 0.3 | | - | 0.3 | | - | 0.5 | 0.2 | - | 0.3 | 0.2 | - | 0.1 | 0.3 | - | 0.1 | 0.2 | - | | 0.3 | - | 0.3 | 0.2 | - | 0.3 | | - | | 0.2 |
| moderate [b] | 0.7 | - | 0.5 | 0.3 | - | 0.3 | 0.3 | - | 0.3 | 0.5 | - | 0.3 | 0.3 | - | | 0.1 | - | 0.2 | 0.1 | - | 0.3 | | - | 0.2 | 0.3 | - | | 0.3 | - | 0.2 | | - | 0.1 |
| extreme [c] | 0.2 | 0.5 | - | | 0.3 | - | | 0.3 | - | 0.2 | 0.3 | - | 0.2 | | - | 0.3 | 0.2 | - | 0.2 | 0.3 | - | 0.3 | 0.2 | - | 0.2 | | - | | 0.2 | - | 0.2 | 0.1 | - |

Note: Superscripts are assigned to the classes. A superscript indicates that the result of the class in this line differs significantly from the class the superscripts belongs to. The second row shows the Cohne's |*d*| values for significant differences between the classes regarding the variable that is labeled in the first line of the first row. The labels of the classes are in the first line of the second row presented by the assigned superscripts. CMS = classroom management strategies, SDO = social dominance orientation, RWA = right-wing authoritarianism.

In summary, the LPA revealed three different attitude profiles. The first attitude profile (class 1) showed a negative rating profile that coincided with the worst ASE and perceived attitude scores, highest perceived stress due to children with ADHD, most frequent use of ineffective CMS, lowest effectiveness rating of effective CMS, and the highest RWA scores compared to the other profiles. The second attitude profile (class 2) was characterized by a moderate rating profile. That profile had the best ASE and perceived attitude score, lowest knowledge about ADHD, most perceived control, lowest perceived stress, highest effectiveness rating of ineffective CMS, lowest stress-reactivity and highest proportion of male participants in comparison to the other profiles. The third attitude profile (class 3) turned out to have an extreme rating profile. Their profile had a more negative perceived attitude than the second did, but a more positive perceived attitude than the first profile. Additionally, this profile was characterized by stress perceived to be moderate, the highest effectiveness rating of effective and the lowest effectiveness rating of ineffective CMS, the lowest SDO and extraversion scores, as well as younger participants compared to the other profiles. Note that this attitude profile added more weight to the positive aspects' rating than to the rating of negative aspects.

## 4. Discussion

A professional's attitude can have a profound impact on children with ADHD, as this attitude can influence the professional's behavior [18,20] which can, in turn, affect the child's severity of ADHD symptoms, their work habits, and social processes [6–9]. Thus, we wanted to identify different attitude profiles and to investigate which ones would prove desirable for professionals working with children with ADHD. We therefore examined the attitudes of teachers in service, in training, as well as PITs, representing groups frequently working with children with ADHD.

### 4.1. Differences between PITs and Teachers

First, we assumed PITs would have a more positive attitude towards children with ADHD than teachers. Our study results support this hypothesis, as the PITs' attitude score measured via the ASE was more positive than the in-service teachers' score. However, the comparison of perceived attitude also revealed that PITs tend to have the most negative score (followed by in-service teachers), differing significantly from pre-service teachers. One explanation for this result might be that PITs are more realistic, have had more actual contacts with children with ADHD, or are more self-critical, especially compared to pre-service teachers. Another reason for this appraisal could be the frame of reference, as the question was not specific in this regard. Participants could on the one hand have referred to inter-individual differences, comparing their attitude with that of others. On the other hand, they could have focused on intra-individual differences by comparing their attitude towards children with ADHD with their attitude towards other children. Correspondingly, PITs might have compared their attitude towards children with ADHD more intra-individually and not taken their probably rather general positive attitude towards "problematic" children that much into account.

### 4.2. Attitude Profiles

Investigating the variable attitude more precisely with an LPA based on the participants response pattern on the ASE's attitude subscales *expectation of positive aspects*, *expectation of negative aspects*, *rating of positive aspects*, and *rating of negative aspects*, we extracted three different attitude profiles that differ mainly in the rating of positive and negative aspects, but not in their expectation. This fact underlines that when examining attitude, it is very important to let participants rate aspects individually and to assess different nuances of positive and negative ratings, as the ASE suggests [36]. Thus, solely asking for (dis-)agreement with statements, as many instruments did before [52,53], will probably ignore existing differences in attitude. Moreover, difference between the identified attitude profiles in knowledge, perceived behavioral control, use and rating of CMS and stress related variables were found. Regarding personality, only SDO, RWA, and extraversion showed a potential influence, though extraversion had the lowest effect. This indicates that only very specific facets of personality

concerning hierarchies and prejudices seem to play a role in this context. Most effect sizes of the differences between the identified attitude profiles were rather small, but can help to characterize the members of the attitude profile classes.

The first attitude profile is characterized by a negative rating style, as both negative and positive aspects tended to be rated rather negatively. This result illustrates that VAS has a possible advantage, as such scales do not reveal a visual division and therefore might facilitate the assessment of a participant's opinion more implicitly than Likert scales, for example. The first attitude profile's tendency to rate negatively is also apparent, considering the effectiveness rating of effective CMS. Participants with the negative rating profile report the lowest effectiveness rating in this regard compared to the other profiles. Furthermore, compared to the others, participants with the negative rating profile report the highest use of ineffective CMS. As these CMS do not help manage children's ADHD symptoms adequately [9,54], it is not surprising that participants with this profile also report the highest perceived stress from children with ADHD compared to the other profiles. This negative experience might facilitate negative expectations, representing one part of their attitude [17,21]. Thus, this profile, compared to the others, accompanies the lowest attitude score measured by the ASE. Participants with the negative rating profile also had the highest RWA scores compared to the others. This finding concurs with the assumption that people with higher RWA scores also have more prejudices against other groups and minorities [55].

The second attitude profile reveals a rather moderate rating style with slightly stronger ratings of negative than of positive aspects. This corresponds with the most strongly perceived control and lowest perceived stress due to children with ADHD compared to the other profiles. However, participants with this profile also have the least knowledge about ADHD, and delivered the highest effectiveness ratings of ineffective CMS compared to the other participants. Furthermore, participants with this profile obtained the highest ASE-measured attitude score, the best self-perceived attitude, and the highest perceived control compared to the other profiles. High-perceived control and low-perceived stress combined with low knowledge could indicate that participants with this profile probably feel that they are able to handle children in general. Another potential interpretation of this profile is that participants with this profile are somewhat uninvolved and do not attach too much value to classroom or therapeutic situations. If so, children with ADHD would not be likely to affect them much—either negatively or positively.

The third attitude profile exhibits rather extreme ratings of positive and negative aspects. Moreover, it places more weight on positive than on negative aspects. Participants with this profile reported the highest effectiveness ratings of effective, and lowest of ineffective CMS compared to the other participants. They also reported moderate perceived stress and a moderate negative attitude towards children with ADHD compared to the other groups. This result may indicate that this profile represents quite realistic participants who are able to evaluate CMS correctly and acknowledge the effort that working with children with ADHD demands. Having the lowest SDO scores compared to the other participants indicates that participants with the extreme rating profile prefer flat to hierarchical structures [39]. The lower extraversion score than those of participants with other attitude profiles reflects this, as extraversion incorporates characteristics such as being dominant or authoritative [56]. This result could imply greater willingness to work with children on an equal level.

*4.3. Desirable Profile for Professionals Working with Children with ADHD*

Now, the question is which of the profiles we identified is most desirable for professionals who work with children with ADHD. As things stand now, we feel compelled to say that it depends on one's perspective. Obviously, the negative rating profile does not seem beneficial for either professional group, as people in this class experience a lot of stress, and children with ADHD have a rather bad reputation in this group. Nevertheless, this profile covered 27% of all in-service teachers and 19% of all pre-service teachers, as well as 17% of all PITs.

The moderate rating profile might be more desirable, as participants with this profile are the only ones with a positive-perceived attitude towards children with ADHD, and they have the best ASE-measured attitude score. As participants with this profile also have a lower stress reactivity compared to the other profiles, they seem to be less stressed by children's (mis-)behavior, the relationship between them and children with ADHD is probably not particularly negative. Moreover, high-perceived behavioral control and low-perceived stress indicate a certain degree of comfort on the part of the professionals. Thus, they might find that being rather relaxed and not very affected by classroom or therapeutic situations is quite comfortable for them and not the worst for children with ADHD. Especially for subject teachers who are less often in contact with their students (or who do not see them for long periods, as in music or art for example), this attitude profile might be favorable. Overall, 27% of all in-service and pre-service teachers and 25% of all PITs were assigned to this profile.

Nevertheless, when focusing on children with ADHD, the extreme rating profile might be preferable, as it is accompanied by more knowledge about ADHD and more correct effectiveness ratings of CMS. The latter appeared to be an important factor influencing the use of effective CMS [25], which again reduces the severity of ADHD symptoms [9]. Combined with the professionals' quite realistic perspective and probably greater involvement, we believe that children with ADHD could profit from this attitude profile. A model analysis also showed that some feeling of stress seems to be conducive to the intention to use effective CMS [27]. Aside from that, the extreme rating profile weighs positive aspects more than negative ones, giving children with ADHD the opportunity to compensate for ADHD-related behavioral problems. The increased probability of using effective CMS and reducing ADHD symptoms can also affect professionals positively. Notably, for teachers in close contact with their students (e.g., main subject or class teachers) this profile's characteristics might be advantageous. Unfortunately, not even half of all these in-service teachers (46%) demonstrated this profile, whereas 54% of all pre-service teachers and 58% of all PITs did.

In summary, our findings highlight the advantage of an LPA over a single examination of attitudes, as the highest attitude score does not necessarily reflect the most beneficial handling of ADHD symptoms.

### 4.4. Implications

The present study findings reveal that nearly a third, a relatively large proportion, of in-service teachers tend to have an attitude profile that can be considered a negative rating profile. This profile has disadvantages for both the professionals themselves and for children with ADHD. Accordingly, it would be important to support professionals with such characteristics in modifying their perspective to improve their collaboration with children with ADHD. Furthermore, it would be interesting if this attitude profile is specific to children with ADHD, or if it applies to children in general. Our working group is currently investigating this point.

Our comparison of the moderate and extreme rating profile reveals the flexibility in handling children with ADHD, as both profiles have their advantages and disadvantages. This fact should be taken into account when trying to convince professionals to implement effective CMS, as some might not perceive a major advantage by doing so. Additionally, this result illustrates the relevance of the professionals' degree of involvement and occupation with and knowledge about the topic of ADHD, and thereby explains why the attitude score itself influences the use of effective CMS only conditionally [25]. Hence, future studies should also assess the degree of involvement. It would also be interesting to discover whether professionals with a moderate rating profile can be encouraged to use effective CMS after knowledge transfer, as they have demonstrated the least knowledge about ADHD to date.

### 4.5. Limitations

A limitation is the small proportion of PITs in the present sample, although compared to the groups' distribution in the general population, it is still rather high. The heterogeneity of the groups

and the demographic differences can also play an important role. In the present study, they were not further analyzed, as they can be seen as fixed and not modifiable.

Second, the extent of experience one can assume when examining pre-service teachers, in-service teachers and PITs is very heterogeneous. To investigate the role of this variable more precisely, it is necessary to assess not just how many years on the job or how many children with ADHD a person has worked with, but also the intensity of those collaborations. That would include the frequency and time period of contacts, and the duration of such contacts in this time period. Future studies should take this into account. Considering contacts of professionals and children with ADHD, the school lessons and therapeutic sessions differ, especially with regard to the number of children that are in one room at the same time. Thus, future studies should focus on one group to examine questions that are more specific. We did so in conducting model analyses for the intention to use effective CMS separately for pre- and in-service teachers [25,27].

Third, online surveys are biased with respect to participants. The dissemination of the link via Facebook groups and email lists reaches only a limited group of people and the truth of the answers cannot be ensured. Besides, professionals with strong negative attitudes towards children with ADHD might not even be willing to participate in such a study. On the other hand, these professionals might be especially keen to take this opportunity to express their opinion. Then again, professionals with a moderate attitude may find an investigation of this topic irrelevant, and would therefore not take part in such a survey. It would thus be beneficial to have a more representative sample—such as all teachers from a selection of schools. Nevertheless, online surveys enable us to collect a large amount of data, and we tried to compensate for any bias by using an attractive incentive.

Addressing the two previous points, it would be interesting to investigate whether differences between professionals working with younger children compared to adolescents can be found. As mentioned above, the current study aimed to provide a first general overview of professionals' attitudes towards children with ADHD. Considering potential age influences of the children on the professionals' attitudes would have been interesting to investigate, though we could not satisfactorily group the different professionals according to children's age, and thus had to refrain from such an analysis as the different range of ages of the children (sub) groups in this sample would have made clear categorization of the children's age difficult. Studies focusing on a specific age group would improve on this.

Furthermore, the present results only enable us to estimate the influence of these various attitude profiles on children with ADHD and their perception thereof. It would be interesting to find out which of these attitude profiles children with ADHD actually favor.

Another limitation of the current study are the relatively low internal consistencies of the subjective norm, Big Five agreeableness, and right wing authoritarianism ratings. This limits the explanatory power of the results. The use of short versions of those scales might be an explanation for these low values, as Cronbach's alpha depends on the length of a scale. This problem is a result of the attempt to make a complex survey with many variables that is still compact, so that not too many participants drop out during completing it.

## 5. Conclusions

The present study examined teachers' and PITs' attitudes towards children with ADHD and identified three different attitude profiles that are characterized by negative, moderate and extreme ratings of expectations towards children with ADHD. It further illustrates that an extreme rating profile might be favorable for children with ADHD, as well as for professionals in close contact with such children. Nevertheless, a moderate rating profile that seems to be related to a somewhat uninvolved mindset might also be of benefit for some professionals. This fact needs to be considered when trying to create a comfortable, more pleasant working environment for children with ADHD and the professionals working with them.

**Supplementary Materials:** The following are available online at http://www.mdpi.com/2071-1050/12/22/9691/s1, BCH-method results z-values.

**Author Contributions:** Conceptualization: M.D., A.E.S. and H.C.; methodology: M.D., A.E.S. and H.C.; validation: H.C., M.S. and A.E.S.; formal analysis: M.D.; investigation: M.D. and A.E.S.; resources: H.C.; data curation: M.D., A.E.S. and H.C.; writing—original draft preparation: M.D.; writing—review and editing: H.C., M.S. and A.E.S.; visualization: M.D.; supervision: H.C. and M.S.; project administration: H.C. and M.S., funding acquisition: H.C. and M.S. All authors have read and agreed to the published version of the manuscript.

**Funding:** This research was part of the project "ADHD in the classroom" that is part of the RTG 2271 and funded by the German Research Foundation (DFG) project number 290878970-GRK 2271, project 1.

**Conflicts of Interest:** The authors declare no conflict of interest.

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
