# Peer review of "Working with Children with ADHD—A Latent Profile Analysis of Teachers’ and Psychotherapists’ Attitudes"

_sustainability, doi:10.3390/su12229691_

Round 1

Reviewer 1 Report

This manuscript reports the results of an online survey study of the attitudes toward students with ADHD among teachers and psychotherapists (most of whom are in training) using a latent profile analysis. Three distinct attitude latent profiles are identified along with variables that are correlated (e.g., right wing authoritarianism, ADHD knowledge, effectiveness ratings of various classroom management strategies [CMS]) with profile group membership. The authors conclude that a profile group that has both positive and negative attitudes toward students with ADHD may be best suited to address the needs of these students in a realistic and knowledgeable fashion.  

There are many strengths to this study and manuscript. Most importantly, this investigation uniquely employs latent profile analysis to identify subgroups of teachers and psychotherapists based on their constellation of attitudes towards students with ADHD.  As stated by the authors, this person-based analytic approach may aid in identifying subgroups of professionals whose work with students with ADHD may be improved in different ways depending on their attitude profile. For example, one of the profile groups (moderate attitudes) appears to be unknowledgeable and perhaps uninterested in ADHD; those professionals may need both educational and motivational supports in order to enhance their work with students. Alternatively the subgroup with primarily negative attitudes may need more support in effective use of evidence-based strategies that are more likely to elicit success than their previous efforts in working with students with ADHD.  The manuscript is clearly written and the findings are straightforward and compelling. Thus, the paper has the potential to provide a unique and substantive contribution to the ADHD research literature.

There also are questions and concerns that need to be addressed in a revised paper:

  1. One variable that is relatively ignored in the introduction and throughout the paper is the potential influence of child age group on teacher and psychotherapist attitudes. That is, the age group (e.g., children vs. adolescents) that professionals work with may influence how they view students with ADHD. The authors need to provide a rationale for why they did not examine this important variable.
  2. The introduction should end with the authors' hypotheses.  Clearly, the authors had specific hypotheses (as alluded to in the discussion) but these were not explicated in the introduction section.
  3. Several of the self-report measures used in this study had relatively low internal consistency estimates including subjective norm rating, Big Five agreeableness, and right wing authoritarianism ratings.  The impact of low internal consistency on the study findings should be addressed and this issue should be identified as a limitation to the study.
  4. Most of the sample were either teachers-in-training or psychotherapists-in-training with a relatively small subgroup of practicing teachers. Thus, the influence of experience in working with students with ADHD is underplayed in this study.  The authors need to address how the makeup of their sample in terms of experience with students impacted their findings.
  5. Similarly, very few of the respondents worked with high school aged students with ADHD. Again, the age group of individuals with whom these professionals have worked is likely to affect their attitudes toward students with ADHD. This needs to be acknowledged more extensively in the document.
  6. The ASE measure will be unfamiliar to many readers of this paper (including me).  Thus, item examples would be helpful in letting readers know what is meant by "attitude toward negative aspects" and "attitude toward positive aspects."  Perhaps a table or appendix could be included that lists the items for this measure.
  7. Effect sizes for the group differences reported on p. 6 should be provided. What was the magnitude of differences between groups on the attitude measures?
  8. Effect sizes also should be reported for differences in auxiliary variables between latent profile groups provided in Table 6. Based on the data in this table, it appears that most of these profile group differences are in the small range (i.e., Cohen's d between 0.2 and 0.3).  The magnitude of these differences should be described in the results section and interpreted in the discussion section as to the implications of relatively small (i.e., subtle) differences between attitude profile groups.

Reviewer 2 Report

The subject of this work seems to me of great interest since the attitude of professionals towards children with ADHD is fundamental for their evolution and cognitive development, and can condition their emotional state. A significant volume of pre-service teacher, in-service teacher and psychotherapist in training has been collected. It is a significant contribution to the global literature on the study on attitudes towards children with ADHD. Good article, much work went into this. Below are some comments / questions and suggestions, personally.

  1. The collection of data through Facebook has important limitations, the first one of identity or profession, I would like to know how they controlled this and what percentage of the information was collected in this way. I think it is an important piece of information considering that the participation had a consideration.
  2. Age data (where there appear to be differences between groups) and professional subgroups (where there also appear to be differences due to age) are presented, but these data or their influence on the results are not analysed.
  3. The personality of the professionals does not seem to have a clear influence on the results, but I have not found the opinion of the authors on this matter.
  4. Regarding the conclusions about the moderate rating profile, I believe that the lower reactivity to stress in this group should be taken into account, in addition to the lower perceived stress and a greater perception of behavioural control. In addition, this group presents an intermediate level in right-wing authoritarianism and social dominance orientation that I believe is necessary to perform the roles of the evaluated professionals, since they are not really egalitarian relationships.
  5. Some things related to the presentation of the information.
  6. Many abbreviations are used in the work (which are not used in the tables) and they do not facilitate the reading of the discussion for its understanding.
  7. Table 5, I think it would make reading easier to point out what the percentages refer to
  8. Table 6, I think there is an error in the data of the significance between groups of the extreme group and perceived stress. In this table the names of the variables are cut.
  9. On line 304 there are two “,”. On line 398, a word appears to be missing where “?? ".

Round 2

Reviewer 1 Report

The authors have responded appropriately to reviewer concerns. The revised manuscript will provide a unique and substantive contribution to the ADHD research literature.